



# Quantifying pollution transport from the Asian monsoon anticyclone into the lower stratosphere

Felix Ploeger[1], Paul Konopka[1], Kaley Walker[2], and Martin Riese[1]

[1]Institute for Energy and Climate Research: Stratosphere (IEK–7), Forschungszentrum Jülich, Jülich, Germany.
[2]Department of Physics, University of Toronto, Toronto, Ontario, Canada.

*Correspondence to:* Felix Ploeger (f.ploeger@fz-juelich.de)

**Abstract.** Pollution transport from the surface to the stratosphere within the Asian monsoon circulation may cause harmful effects on stratospheric chemistry and climate. Here, we investigate air mass transport from the monsoon anticyclone into the stratosphere using a Lagrangian chemistry transport model. We show how two main transport pathways from the anticyclone emerge: (i) into the tropical stratosphere (tropical pipe), and (ii) into the Northern hemisphere (NH) extra-tropical lower strato-

5  sphere. Maximum anticyclone air mass fractions reach around 5% in the tropical pipe and 15% in the extra-tropical lowermost stratosphere over the course of a year. The anticyclone air mass fraction correlates well with satellite hydrogen cyanide (HCN) and carbon monoxide (CO) observations, corroborating that pollution is transported deep into the tropical stratosphere from the Asian monsoon anticyclone. Cross-tropopause transport occurs in a vertical chimney, but with the emissions transported quasi-horizontally along isentropes above the tropopause into the tropics and NH.

## 1  Introduction

The Asian summer monsoon circulation provides a pathway for anthropogenic pollution into the stratosphere (e.g., *Randel et al.*, 2010), where it may crucially affect stratospheric chemistry and radiation. A related phenomenon is the build-up of the Asian tropopause aerosol layer ATAL (*Vernier et al.*, 2011), which has recently been estimated to cause a significant regional radiative forcing of -0.1 W/m$^2$ (*Vernier et al.*, 2015), cooling the Earth's surface. Hence, transport in the Asian monsoon is likely an

15  important factor for climate change.

Transport by the Asian monsoon includes convection over the Bay of Bengal, Northern India and the South China Sea (e.g., *Tzella and Legras*, 2011; *Wright et al.*, 2011; *Bergman et al.*, 2012). At higher levels monsoon transport is dominated by a strong anticyclonic circulation (*Randel and Park*, 2006) with confinement and slow uplift of air in the upper troposphere and lower stratosphere UTLS (e.g., *Park et al.*, 2009). Related to this transport are positive anomalies of trace gases with

20  tropospheric sources and negative anomalies of trace gases with stratospheric sources (e.g., *Park et al.*, 2008). The detailed upward transport from the convective outflow to higher levels involves a vertical conduit over the southern Tibetan plateau (*Bergman et al.*, 2013). In addition, convective uplift by typhoons has been shown to inject air masses into the outer region of the anticyclonic circulation (*Vogel et al.*, 2014). The interplay of these processes results in fast upward transport into the lower





stratosphere and an enhanced fraction of young air in the monsoon UTLS region (*Ploeger and Birner*, 2016). Convection over land causes particularly fast upward transport (*Tissier and Legras*, 2016).

Based on global satellite observations of hydrogen cyanide (HCN), *Randel et al.* (2010) argued that upward transport from the Asian monsoon reaches deep into the tropical stratosphere. Water vapour observations and simulations, on the other hand,

show transport from the monsoon anticyclone mainly into the extra-tropical lower stratosphere (e.g., *Dethof et al.*, 1999). As stratospheric water vapour is strongly controlled by cold temperatures around the tropopause these results are not necessarily contrary. However, even tracer independent model diagnostics yielded controversial results, recently. On the one hand, the back trajectory study of *Garny and Randel* (2016) shows strongest transport from the anticyclone directly into the tropical stratosphere. On the other hand, climate model simulations by *Orbe et al.* (2015) show the tropopause crossing of air masses

from the anticyclone largely in the extra-tropics and subsequent transport into the extra-tropical lower stratosphere.

Here, we use tracer independent model diagnostics (i.e., independent from species' chemistry and emissions) in combination with satellite observations of the tropospheric tracers HCN and CO to investigate the pathways of pollution from the Asian monsoon anticyclone to the lower stratosphere, and quantify the related amount of air originating in the monsoon anticyclone. First, we demonstrate how transport from the anticyclone can be divided into two main pathways directing into (i) the tropical

pipe and (ii) the Northern hemisphere (NH) extra-tropical lowermost stratosphere, over the course of a year following the monsoon season. Second, we discuss the detailed transport across the tropopause in the monsoon, and argue that the Asian monsoon acts as a vertical "chimney" with strong horizontal transport ("blowing") on top for air mass transport into the stratosphere (see Sect. 4).

## 2   Method

We quantify air mass transport from the Asian monsoon anticyclone using simulations with the Lagrangian chemistry transport model CLaMS (*McKenna et al.*, 2002; *Konopka et al.*, 2004; *Pommrich et al.*, 2014). CLaMS uses an isentropic vertical coordinate throughout the UTLS, and the model transport is driven with horizontal winds and total diabatic heating rates from European Centre of Medium-Range Weather Forecasts (ECMWF) ERA-Interim reanalysis (*Dee et al.*, 2011). We included an air-mass origin tracer in the model to diagnose the fraction of air at any location in the stratosphere which has left the

Asian monsoon anticyclone during the previous monsoon season. In addition we consider carbon monoxide (CO), with the CO lower boundary in CLaMS derived from Atmospheric Infra-red Sounder (AIRS) version 6 satellite measurements following the method described in *Pommrich et al.* (2014), and relevant chemistry for the UTLS region included (*Pommrich et al.*, 2014).

It has recently been shown that trace gas confinement within the monsoon anticyclone core can be best described by potential vorticity (PV) contours (*Garny and Randel*, 2013), and that the anticyclone core can be clearly distinguished from the

surrounding atmosphere in a layer around 380 K potential temperature (*Ploeger et al.*, 2015; *Ungermann et al.*, 2016). We therefore apply the method of *Ploeger et al.* (2015) to determine the PV-value related to the anticyclone border from the maximum PV-gradient on every day during summers 2010–2013 at the 370 and 380 K potential temperature surfaces (see appendix for further details, and the online supplement for the data).


The anticyclone tracer is initialized inside the PV-contour enclosing the anticyclone core in the 370–380 K layer on each day during July–August of the years 2010–2013, and is advected as an inert tracer during the following year. On 1 July of the year thereafter, the tracer is set to zero everywhere and is then reinitialized for the following monsoon season. By definition, the tracer mixing ratio at any location in the stratosphere equals the fraction of air which has left the monsoon anticyclone

during the previous monsoon season (compare *Orbe et al.*, 2013). Initializing the air mass origin tracer in the UTLS part of the Asian monsoon avoids our results to be affected by small-scale transport processes in the troposphere (e.g., convection), whose representation in global reanalysis data is uncertain (e.g., *Russo et al.*, 2011). This choice of method is suitable to study the transport of air from the anticyclone, irrespective of where it originated at the surface. The impact of different boundary layer source regions on the Asian monsoon UTLS is an important research topic itself (e.g., *Vogel et al.*, 2015; *Tissier and Legras*,

2016).

The anticyclone air mass tracer is compared to global HCN measurements from the Atmospheric Chemistry Experiment Fourier Transform Spectrometer (ACE-FTS) satellite instrument (*Bernath et al.*, 2005). This data has been presented and discussed by *Randel et al.* (2010) to be a valid tracer for Asian monsoon pollution. For the results of this paper we use HCN from the updated ACE–FTS level 2 data version 3.5 (*Boone et al.*, 2005, 2013) during the period between 1 July 2010 and 30

June 2014, which is in good agreement with the results shown by *Randel et al.* (2010). Physically unrealistic outliers in the ACE–FTS data have been filtered out following *Sheese et al.* (2015), discarding data with quality flag greater than 3. Furthermore, we use CO observations from the Microwave Limb Sounder (MLS) on board the Aura satellite (*Pumphrey et al.*, 2007; *Livesey et al.*, 2008), for validating Asian monsoon transport in the model simulation.

## 3  Results

Figure 1 presents the anticyclone air mass fraction and compares with HCN satellite observations from ACE–FTS. During July–September (Fig. 1a) the anticyclone air is transported into the lower stratosphere mainly in the subtropics (between 20°–40°N). During fall (October–December, Fig. 1b), the anticyclone air disperses throughout the NH lower stratosphere, even reaching the tropics and Southern hemisphere (SH). Strong wintertime tropical upwelling related to the stratospheric Brewer-Dobson circulation lifts the anticyclone air in the tropics during the following winter (Fig. 1c). Related downwelling in the

extra-tropics flushes the anticyclone air out of the NH lower stratosphere. During spring (Fig. 1d), the anticyclone air in the tropical pipe rises further while the extra-tropical lower stratosphere is cleaned. Hence, two main pathways emerge for air from the Asian monsoon anticyclone into the stratosphere. First, a fast transport pathway is directed into the NH extra-tropical stratosphere (extra-tropical pathway). Second, a slower pathway is directed into the tropical stratosphere and deep into the stratosphere related to ascent within the tropical pipe (tropical pathway).

Contours of ACE–FTS measured HCN show that the simulated anticyclone air mass fraction correlates well with satellite-observed pollution (for a discussion of this data as a tracer for pollution from the Asian monsoon compare *Randel et al.*, 2010). In analogy to the model tracer, observed HCN peaks in the subtropical and extra-tropical lower stratosphere during and directly following the monsoon season (Figs. 1a, b). Further, HCN shows a secondary maximum in the tropical pipe. Hence, enhanced





HCN in the tropical pipe, which was interpreted by *Randel et al.* (2010) to originate in the monsoon, correlates well with the air mass signal from the anticyclone. The fact that the ascending tropical HCN signal slightly lags the model tracer signal (Fig. 1d) is consistent with the overestimated tropical upwelling in ERA-Interim (e.g., *Dee et al.*, 2011).

However, HCN exhibits enhanced concentrations also in the SH subtropics during austral spring to summer (Fig. 1b, c), consistent with independent satellite observations from the Michelson Interferometer for Passive Atmospheric Sounding MI-PAS (*Glatthor et al.*, 2015). Hence, a contribution from the SH to stratospheric HCN can not be ruled out. Furthermore, the irregular tape-recorder signal in the deseasonalized anomaly of tropical HCN during 2005–2008 (*Pumphrey et al.*, 2007) has been linked to irregularly occurring biomass burning in Indonesia (*Pommrich et al.*, 2010). Compared to these studies, the focus here is on the annually repeating seasonal signal discussed by *Randel et al.* (2010). The qualitative agreement between the transport pathways of HCN and air mass from the monsoon indicates that transport from the Asian monsoon anticyclone has the potential to significantly contribute to the annual signal in HCN concentrations in the stratosphere. In the following, we focus on air mass transport from the Asian monsoon, which clearly reaches the tropical pipe (Figs. 1) and therefore may cause substantial pollution transport deep into the stratosphere.

The time series of air mass fractions in Fig. 2 show that the amount of anticyclone air peaks in the NH extra-tropical lowermost stratosphere in October, reaching around 15% at 380 K. In the tropics at 460 K (the lower edge of the tropical pipe) the amount of anticyclone air peaks in December, reaching around 5%. This later timing of the peak in the tropics compared to the extra-tropics is related to the higher potential temperature level (460 vs. 380 K) and slow tropical upwelling. At lower levels (here 400 K, red dashed) the tropical anticyclone air mass fraction peaks earlier, around October. While the anticyclone air fraction in the extra-tropical stratosphere peaks with a value that is more than twice as high compared to the tropical anticyclone air fraction, the anticyclone air transported to the tropics remains much longer in the stratosphere and exceeds the extra-tropical amount after about half a year. The large standard deviation around the extra-tropical zonal mean value (grey shading in Fig. 2) indicates strong variability in the extra-tropical lowermost stratosphere tracer distribution, related to various processes (e.g., Rossby-wave breaking). At the lower end of the tropical pipe (460 K), the tracer distribution is more homogeneous as reflected in a smaller standard deviation.

To further understand the details of transport from the monsoon anticyclone into the stratosphere we investigate the direction of tropopause crossing. Recently, the question was raised whether the air confined within the monsoon anticyclone crosses the tropopause vertically or horizontally or, in other words, whether the monsoon acts mainly as a vertical "chimney" or as an isentropic "blower" for cross-tropopause transport (*Pan et al.*, 2016). The good agreement of carbon monoxide distributions in the monsoon region at 380 K between the CLaMS simulation and MLS satellite observations shows that the model reliably simulates transport in the monsoon anticyclone (Fig. 3a, b upper panels). Note that the figure shows the deviation of CO from the zonal mean to emphasize the anomalous character of monsoon transport. In particular, the positive CO anomaly in the monsoon agrees well between model and observations, and even the weak positive anomalies to the north-west and north-east of the monsoon indicating regions of frequent eddy shedding (*Hsu and Plumb*, 2001; *Popovic and Plumb*, 2001).

In order to clearly separate tropospheric and stratospheric air we transform the data to a tropopause-based vertical coordinate, chosen as the distance to the local tropopause in potential temperature before calculating all averages (compare e.g.,





*Birner et al.*, 2002; *Hoor et al.*, 2004, for using this method in a different context). The distributions in the monsoon region change remarkably when viewed in tropopause-based coordinates along a surface at 10 K above the local tropopause (Fig. 3a, b lower panels). The positive CO anomaly significantly weakens, as an effect of the averaging procedure following the tropopause, indicating that a considerable part of the trace gas anomaly in the monsoon is related to the upward bulging

tropopause in the monsoon region. However, the fact that parts of the anomaly remain indicates upward transport across the tropopause above the monsoon. Also for the tropopause-based map, CO distributions from CLaMS and MLS observations agree reliably well in the monsoon region. Significant differences between CLaMS and MLS exist only at mid-latitudes (already observed by *Pommrich et al.*, 2014) and above the West Pacific and Maritime continent.

Figure 3c shows analogous maps for the anticyclone air mass fraction. Again, tropopause based averaging weakens the
positive monsoon anomaly. However, a clear positive anomaly remains centred in the monsoon region above the tropopause. This indicates that cross-tropopause transport into the stratosphere in the monsoon occurs to a large degree in the vertical direction. Vertical transport, diagnosed from the ERA-Interim total diabatic heating rate, is consistent with this finding showing maximum upward velocity in the anticyclone (grey contours in Fig. 3). The stronger degradation of the monsoon anomaly for CO as compared to the inert air mass tracer is related to the finite (∼4 months) lifetime of CO. As a consequence, CO mixing
ratios degrade rapidly at levels around the tropopause where vertical transport is slow.

An unambiguous picture of air mass transport across the tropopause can only be deduced from the inert air mass origin tracer in the model. Figure 4 shows the anticyclone air mass fraction averaged over the zonal section of the Asian monsoon (40°–100°E) and over periods of about a week (with all averages carried out in tropopause-based coordinates). Directly after the main monsoon season at the end of August (Fig. 4a) the largest amount of anticyclone air is located in the subtropics between 20°–
40°N around and above the tropopause. One month later, this air has been further transported upwards and resides clearly above the tropopause (Fig. 4b). Hence, cross-tropopause transport of anticyclone air occurs mainly vertically across the subtropical tropopause, like in a chimney. Above the tropopause, however, in a layer between about 380–430 K the air from the anticyclone is strongly affected by horizontal transport processes and is largely mixed into the NH extra-tropics and into the tropics (Fig. 4b–d). Strong horizontal transport above about 380 K in NH summer and fall is likely related to enhanced subtropical Rossby-
wave breaking during this season (see *Homeyer and Bowman*, 2012). Fastest uplift in the subtropics is consistent with largest upward velocity in that region (black contours in Fig. 4a). Note that ERA-Interim cross-isentropic vertical velocities in August show even downwelling equator wards of about 10°N in the 380–410 K layer.

## 4  Discussion

There has been a controversial recent scientific debate on if and how the air masses from the Asian monsoon anticyclone reach
the lower stratosphere. *Garny and Randel* (2016) concluded from 60 day backward trajectory ensembles that the largest fraction of anticyclone air reaches the tropical stratosphere via a direct pathway. *Orbe et al.* (2015) analysed air mass origin tracers in a climate model. They found that Asian surface air is transported upwards in the monsoon, reaches the extra-tropical tropopause within a few days, and is first transported quasi-horizontally into the extra-tropical lower stratosphere before eventually being





transported subsequently into the tropics. A very recent study by *Pan et al.* (2016) also shows mainly quasi-horizontal isentropic transport out of the monsoon anticyclone into the lower stratosphere.

Here, we focus on transport from the anticyclone deep into the stratosphere. Using a PV-gradient based definition of the anticyclone edge, we trace the anticyclone air over an entire year following the monsoon season. Our analysis shows that the air from the anticyclone crosses the subtropical tropopause vertically (here cross-isentropical) and is subsequently transported horizontally (along isentropes) in the stratosphere to both the tropics and to NH extra-tropics, as illustrated in Fig. 5. The vertical nature of cross-tropopause transport is consistent with the findings of *Garny and Randel* (2016), but with the addition that above the tropopause a substantial amount of anticyclone air is mixed into the NH extra-tropics. This strong horizontal transport is, on the other hand, consistent with *Orbe et al.* (2015) and *Pan et al.* (2016), but with the difference that horizontal transport in our case occurs mainly above the tropopause. It is important to note that we defined vertical and horizontal transport with respect to potential temperature as the vertical coordinate. Therefore, horizontal transport can be directly interpreted as isentropical mixing.

Hence, in summary we refine the findings of *Orbe et al.* (2015), *Garny and Randel* (2016) and *Pan et al.* (2016) by describing transport from the Asian monsoon anticyclone into the stratosphere as a "blowing chimney", using the terminology of *Pan et al.* (2016). This characterization emphasizes the vertical "chimney–like" nature of cross-tropopause transport (with respect to potential temperature as vertical coordinate), but with the emissions transported away quasi-horizontally along isentropes above the tropopause (see Fig. 5). This quasi-horizontal transport pathway from the monsoon into the UTLS is supported by recent in-situ measurements (*Mueller et al.*, 2016). At lower levels below the tropical tropopause (about 380 K) horizontal transport from the anticyclone core to the NH extra-tropics is very weak due to strong gradients in PV, in agreement with the findings of *Garny and Randel* (2016). The chimney-like nature of cross-tropopause transport is likely related to the existence of a vertical transport conduit found by *Bergman et al.* (2013).

So far, our conclusions concern air masses from the anticyclone core. To investigate differences in transport from the anticyclone edge, we initialized an anticyclone edge tracer in CLaMS (between PV-contours of the anticyclone border PV$^*$ and PV$^* + 2$ PVU, see appendix), whose mixing ratio by definition yields the fraction of air originating from the anticyclone edge during the last monsoon season. Figures 4e and f show the air mass fraction from the anticyclone edge at the end of August and at the end of November. Comparison to the air mass fraction from the anticyclone core shows that directly after the monsoon season (Figs. 4a, e) air from the anticyclone edge is transported faster in the horizontal direction into the tropics and into NH extra-tropics. This is a consequence of air masses in the anticyclone edge region being less well confined as compared to air masses in the anticyclone core. After a few months, however, the two distributions of anticyclone edge and core air align (Figs. 4d, f), showing that on the long-term air masses ascending in the anticyclone core and air masses injected into the anticyclone edge (e.g., by typhoons, see *Vogel et al.*, 2014) follow the same transport pathways. The higher fraction of air from the anticyclone edge compared to the core is likely a result of the larger area of the edge region. Note that air masses in the anticyclone edge may have originated in the anticyclone core at lower levels, as suggested by the vertical transport conduit pathway (compare *Bergman et al.*, 2013).



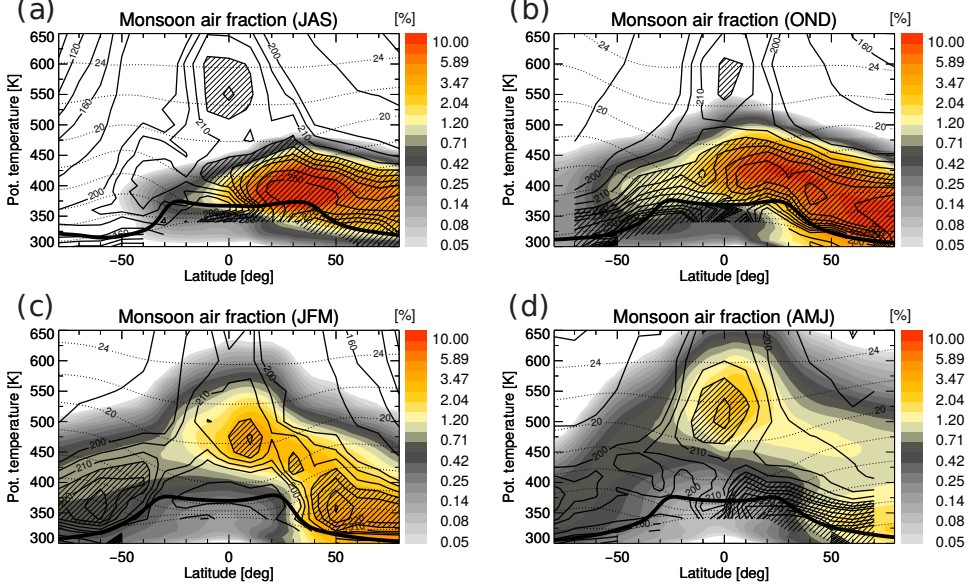

**Figure 1.** Seasonal evolution of climatological (2010–2013) zonal mean monsoon air mass fraction from CLaMS (colour-coded) and HCN from ACE–FTS observations (black contours) during July–September (a), October–December (b), January–March (c), and April–June (d). Regions with HCN values above 215 pptv are hatched. The thick black line shows the (WMO) tropopause, thin black lines show altitude levels (2 km spacing).

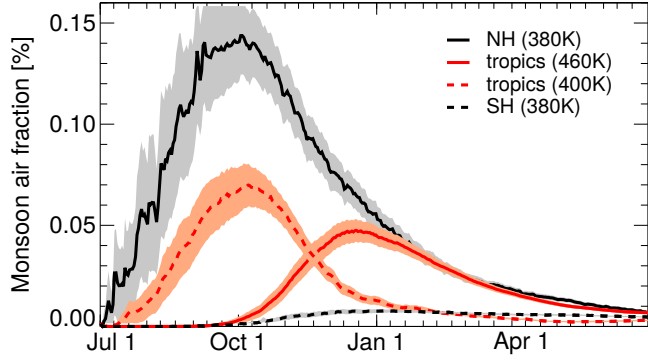

**Figure 2.** Climatological time series of monsoon air mass fraction in the tropical lower stratosphere (15°S–15°N) at 450 K (red solid) and 400 K (red dashed), and in the extra-tropical lower stratosphere (50°–70°N/S) at 380 K in the NH (grey solid) and in the SH (grey dashed). Shading shows the mean standard deviation for the zonal average (multiplied by 0.25, for better visibility), as a measure of variability.





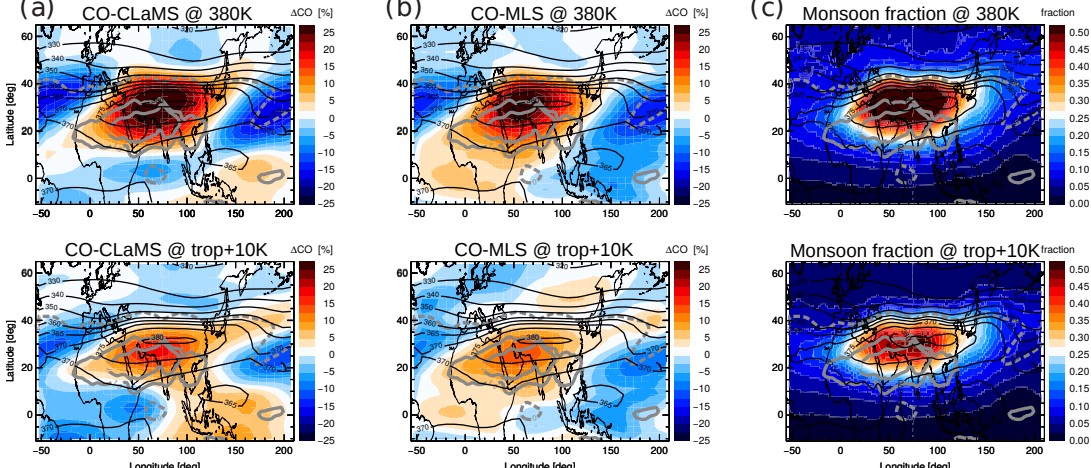

**Figure 3.** Maps of (a) carbon monoxide from CLaMS simulation, (b) CO from MLS satellite observations, and (c) monsoon air mass fraction from CLaMS, all for July–September. Top panels show maps at 380 K potential temperature, bottom panels show maps along a surface at 10 K above the local (WMO) tropopause. For CO the deviation from the zonal mean is shown in percent (ΔCO). Black contours show the potential temperature of the local WMO tropopause, grey contours cross-isentropic (diabatic) vertical velocity $d\theta/dt$ (solid: 1, 1.3 K/day; dashed: 0 K/day). CO climatologies were calculated for the period 2004–2016, air mass fraction climatologies for 2010–2013.

## 5  Conclusions

The anticyclone air fraction of 5% in the tropical pipe appears small if compared to the 15% fraction in the NH extra-tropical lowermost stratosphere. However, as tropical air ascends deep into the stratosphere with the rising branch of the Brewer-Dobson circulation while extra-tropical air is flushed out of the stratosphere within a few months, the impact of this tropical
5  anticyclone air on stratospheric chemistry and climate may be substantial. Our model simulation shows that the tropical anticyclone air correlates well with the annual cycle in satellite observed HCN over the course of a year and hence likely causes pollution transport deep into the stratosphere and contributes to the stratospheric aerosol loading. Therefore, changes in these two pathways of pollution from the Asian monsoon anticyclone into the stratosphere likely affect chemistry and radiation and may be important for causing feedback effects in a changing climate.

## 10  Appendix A:  Asian monsoon anticyclone border from PV-gradient

To separate the Asian monsoon anticyclone core region from its surroundings we follow the method of *Ploeger et al.* (2015). This method is based on the existence of an enhanced PV gradient indicating a transport barrier between the core and the surrounding region, similar but weaker to the polar vortex edge (compare e.g., *Nash et al.*, 1996). The anticyclone core is defined as the region enclosed by the PV–contour PV* corresponding to the maximum gradient of PV with respect to a monsoon-
15  centred equivalent latitude (*Ploeger et al.*, 2015). Note that the PV field has to be smoothed by averaging over a time-window





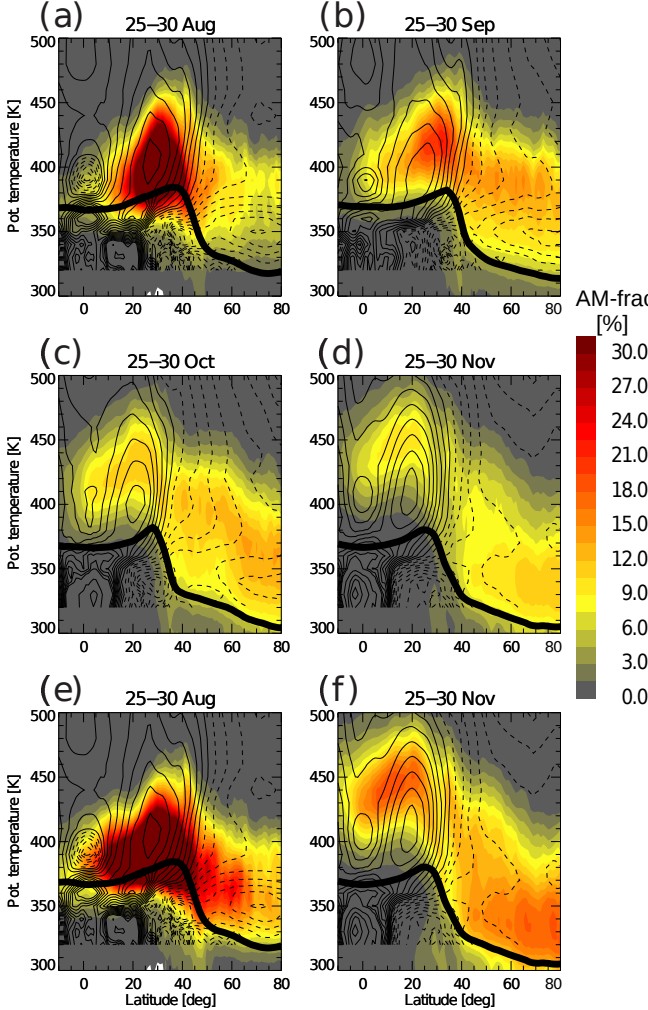

**Figure 4.** Latitude section of monsoon air mass fraction averaged over longitudes between 40–100°E for the (climatological 2010–2013) periods 25–30 August (a), September (b), October (c), and November (d). The averaging has been carried out in tropopause-based vertical coordinates, and the data has afterwards been adjusted vertically for plotting by adding the mean tropopause potential temperature (grey line). (e, f) Same as (a, d) but for the monsoon edge fraction, calculated from the monsoon edge tracer (see text). Thin black contours show total diabatic vertical velocity $d\theta/dt$ (positive values solid, negative values dashed, contour-spacing 0.2 K/day), the thick black line shows the mean tropopause.

around the given date before the calculation, for a clear gradient maximum to emerge, due to strong dynamic variability of the monsoon circulation. The situation for the 6 July 2011 at 380 K is illustrated in (Fig. 6a), showing the time-averaged PV field (averaged over 5–7 July 2011), with the anticyclone core (region of lowest PV) enclosed by the deduced transport barrier (thick black line).





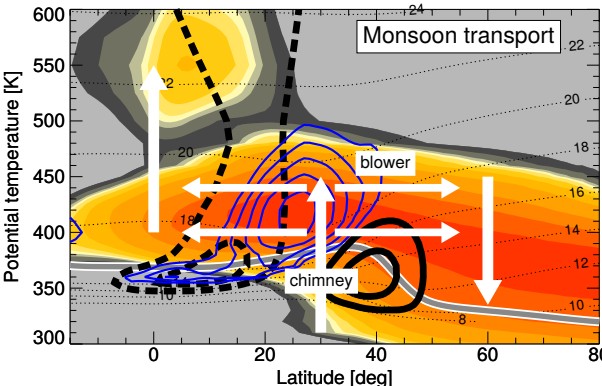

**Figure 5.** Schematic of pollution transport from the monsoon into the stratosphere. Colour shading shows zonal mean anticyclone air mass fraction (October below about 500 K, June above), with white arrows illustrating the dominant transport pathways. Black contours show zonal wind ($\pm 15$, 25 u/s solid/dashed), blue contours diabatic heating rates (from 1 K/day increasing in 0.2 K/day steps), thin black geopotential height, thick grey line the (WMO) tropopause, all from ERA–Interim for June–August and zonally averaged over the monsoon region (40–100°E).

The calculation yields a well-defined PV-value for most days of the summers 2010–2013 (red symbols in Fig. 6b, c). Missing data in the time series of the anticyclone border of each summer have been filled in by linear interpolation (black symbols) in time from the neighbouring values. At days before the first day (and after the last day) when the PV-gradient criterion holds no anticyclone border PV-value has been estimated (no extrapolation), and the time series ends. This procedure results

in a smooth PV-time series of the anticyclone border during July–August (Fig. 6b, c). The model tracer has been initialized with unity within the anticyclone core in the 370–380 K layer during July–August. Note that we used the time-averaged PV field for the initialization criterion. Both the anticyclone border PV-value $PV^*$ and the time-averaged PV-field calculated from ERA-Interim for the summers 2010–2013 are available from the online supplement. The tracer mixing ratio, by definition, yields the mass fraction of air from the anticyclone core region during the previous monsoon season (compare Sect. 2, and e.g.,

*Orbe et al.*, 2013). In analogy, the anticyclone edge tracer is initialized with unity between PV-contours of the anticyclone border $PV^*$ and $PV^* + 2\,PVU$ (see Fig. 6a), providing the mass fraction of air from the anticyclone edge region during the previous monsoon season.

*Acknowledgements.* We thank Rolf Müller, Bärbel Vogel, Jens-Uwe Grooß and Laura Pan for helpful discussion. We further thank the ECMWF for providing reanalysis data, the MLS team for providing CO satellite observations, and the ACE–FTS team for providing HCN

satellite observations. The Atmospheric Chemistry Experiment (ACE), also known as SCISAT, is a Canadian-led mission mainly supported by the Canadian Space Agency and the Natural Sciences and Engineering Research Council of Canada. The CLaMS model data may be requested from the corresponding author (f.ploeger@fz-juelich.de). The PV-barrier time series for the years 2010–2013 is available from the online supplement. The ACE-FTS Level 2 data used in this study can be obtained via the ACE-FTS website, http://www.ace.uwaterloo.ca.





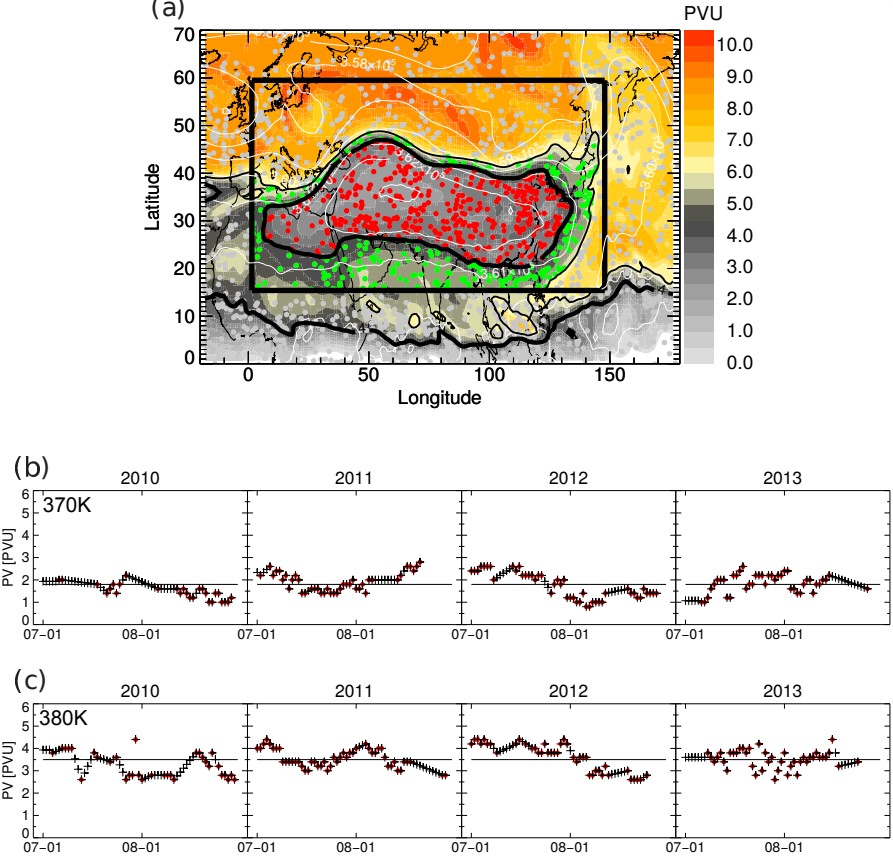

**Figure 6.** (a) Map of time-averaged PV field at 380 K on 6 July 2011, calculated as the average over the PV distribution for 5–7 July 2011. The thick black contour shows the calculated PV-gradient based anticyclone border PV$^*$ (4 PVU for that date), the thin black contour shows PV$^*$+2 PVU. Thin white contours show selected Montgomery stream function values. Filled circles show CLaMS air parcels between 379–380 K, with parcels inside the anticyclone core coloured red, at the anticyclone edge coloured green. The black rectangle indicates the regional restriction of the calculation (see text). (b) Time-series of PV-gradient based anticyclone border PV-value at 370 K, with the calculated barrier as red circles and interpolated barrier (at days where the calculation did not work, interpolated from existing neighbour values) as black crosses. (c) Same as (b) but for 380 K.

This work was supported by the HGF Young Investigators Group A–SPECi ("Assessment of Stratospheric Processes and their Effects on Climate variability").





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
