# Peer review of "Quantifying pollution transport from the Asian monsoon anticyclone into the lower stratosphere"

_Atmospheric Chemistry and Physics, 2017_

## Referee Comment (RC1) · Anonymous Referee #1 · 4 Mar 2017

The paper explores the transport of air inside the Asian monsoon anticyclone into the stratosphere using a Lagrangian chemistry transport model. Some results are compared to satellite observations of HCN from ACE-FTS and CO from MLS. The results suggest two pathways of transport from the core of the anticyclone into the extratropical and tropical stratosphere. The transport of pollutants into the stratosphere is a topic of high relevance, the methods and the data employed in the paper are adequate, and the paper is nicely written and easy to follow. However, there are a few fundamental issues that should be addressed before the paper is published, listed below.

*General comments*

1) The authors argue that they identify a vertical pathway across the tropopause. However, since the initialization of the origin tracer (370-380 K) is not explicitly limited to the troposphere, the derived conclusions regarding cross-tropopause transport are not rigorous. The authors should discuss the position of the initialized tracer relative to the tropopause and analyze the sensitivity of their results to the initial location chosen.

2) It would notably improve the paper if a figure similar to Figure 4 but for CO concentrations were included, comparing the model to the observations. Alternatively, a figure similar to the lower panels of Fig. 3 extended to show CO and the origin tracer on additional levels relative to the tropopause (e.g. up to 50 K above the tropopause). Any of these suggested figures would highlight the differences and similarities between CO and the origin tracer (are they due only to the CO lifetime?), provide further confirmation of the correct representation of transport in the model, and facilitate the comparison of the results with previous works.

3) Figure 2 calls for a more detailed explanation of the large variability in the time series (divided by a factor of 4 in the Figure). In particular, it is unclear to the reader how the standard deviations are defined (with respect to the various years or for each longitude point within each latitudinal band?). This is an important point, because the validity of the conclusions on the seasonality is questioned given the large variability around the time series, and thus it needs to be clarified exactly what the spread represents.

*Minor comments*

P3L11-18: A description of the data the temporal and spatial resolution should be included, to help the reader understand how the data is employed for the model
comparisons. For example, ACE-FTS data does not allow high-temporally resolved comparison as that in Fig. 4.

P3L25: In Fig. 1d (AMJ) there is little agreement between the model and the observations in the NH. Could you comment on why this might be the case?

P3L26: In Fig. 1a there is a maximum of HCN in the tropical pipe, but no corresponding maximum in the origin tracer. Is this associated with the ascent from the year before?

P3L2-3: It is not clear what you refer to. Could you refer to specific contours?

P3L15: How do you define the lower edge of the tropical pipe?

P3L20: I don't see in Fig. 2 that "the anticyclone air exceeds the extra-tropical amount by half a year". Do you mean that from February until the next summer the tropical and extra-tropical monsoon air mass fractions are approximately equal?

P4L21-24: As mentioned in the major comment 3), a clarification on how the variability is computed and what it implies for the conclusions on the seasonality is needed.

P5L11: "cross-tropopause transport into the stratosphere in the monsoon occurs to a large degree in the vertical direction". You could actually quantify to what degree this happens. This is in line with major comment 2), suggesting a more detailed analysis of CO behavior relative to the tropopause, in the model and the observations.

P6L20: How is this related to the vertical conduit from the boundary layer?

*Technical corrections/suggestions*

P1L9: emissions → pollutants

P2L11: tracer-independent

P2L16: "and argue that in terms of air mass transport into the stratosphere, the Asian monsoon..."

P2L23-26: I would recommend introducing the tracer here rather than at the Appendix. Specifically mention that it is defined as unity inside the monsoon anticyclone.

P2L31-32: Here and throughout the paper, PV-value and PV-gradient do not need a

**[ACPD](...)**
hyphen.

Fig. 1: Perhaps the hatching could be removed, as it is hard to distinguish the colors underneath.

Fig. 2: There is a typo either in the legend or in the figure caption, is the level 450 Kor 460 K? Also, the lines are not grey as the caption says, but black.

P5L9: tropopause-based

P5L29: Is it really controversial or just a scientific debate?

P6L10: This does not discard isentropic advection

P6L16: emissions → pollutants

Figure 5: Please add a colorbar.

---

## Referee Comment (RC2) · Anonymous Referee #2 · 27 Mar 2017

Review of "Quantifying pollution transport from the Asian monsoon anticyclone into the lower stratosphere" by F.Ploeger et al.

The analyses of Ploeger et al., analyses the transport pathways from the Asian summer monsoon into the stratosphere using the Chemical Lagrangian Model of the Stratosphere (CLaMS). The CLaMS model is driven by ERA Interim climatologies and provides information on the transport of CO which is initialized by AIRS CO data at the lower boundary. In addition an artificial air mass origin tracer for the anticyclone is included into the model to provide information on the export of anticyclone air masses. This tracer is initialized in the monsoon upper troposphere between 370K-380K enclosed by a PV contour which represents the anticyclone. Importantly, the authors apply a tropopause based analysis, which avoids artificial cross tropopause transport signals from averaging over different tropopause heights (i.e. potential temperatures

at the tropopause) and which also allows to distinguish between cross diabatic vertical transport and quasihorizontal transport.The authors show that vertical (cross isentropic) transport across the tropopause in the monsoon region is of high relevance and subsequent quasi-horizontal transport occurs to lower and higher latitudes. They further differentiate between the monsoon edge and the core and find that quasihorizontal transport from the monsoon edge on the short term is faster than transport from the core, but that both source regions follow the same transport pathways to the tropics or extratropics on the longer time scale. Particularly they quantify the potential contributions to the deeper branch of the Brewer-Dobson circulation and the shallow branch to be 5% and 15 %, respectively.

Currently a debate about the preferred pathways for pollution export from the anticyclone is going on. Thus the paper adds to this discussion and provides a very valuable insight into the potential pathways and partly mechanisms and thus is of high relevance. It quantifies the fractions of monsoon air, which are exported from the monsoon to the tropics as well as to the extratropics. The paper is well structured, fluently written and the methods are sound and I judge it as an excellent contribution to the field.

I recommend this paper almost as it is.

Minor remarks: p.5, l.31: What is meant with 'direct pathway'? I guess quasihorizontal roughly along isentropes from 380K-400K into the tropical tropopause region? p.5, l.33: Similarly: "....transported upwards in the monsoon and reaches the extratropical tropopause within a few days...": The monsoon tropopause is meant here with extratropical, please clarify. p.5, l.27: equator wards -> equatorwards

---

## Author Comment (AC1) · 2 May 2017

We thank the Reviewer for her/his careful consideration of the manuscript and her/his well thought-out comments. These certainly helped to improve the paper. In the following, we address all comments and questions raised (Reviewer's comments in italics). Text changes in the manuscript are highlighted in color (except minor wording changes).

General comments:

*1) The authors argue that they identify a vertical pathway across the tropopause. However, since the initialization of the origin tracer (370-380 K) is not explicitly limited to the troposphere, the derived conclusions regarding cross-tropopause transport are not rigorous. The authors should discuss the position of the initialized tracer relative to the*

[Figure]

*tropopause and analyze the sensitivity of their results to the initial location chosen.*

The initialization of the anticyclone air mass tracer is indeed a critical point, as the Reviewer points out. However, a clear distinction between anticyclone core and surroundings based on an enhanced PV-gradient is only possible in a layer around 380K potential temperature (see Ploeger et al., 2015). Therefore, based on the PV-identification of the anticyclone the tracer can only be initialized between 370–380K.

In the revised version of the manuscript we analyse the frequency of occurrence of tropopause heights in the anticyclone lower than 380K (see new Fig. 7 and the discussion in "Method" and "Appendix"). It turns out that for less than 4located below 380K. Hence, the initialization region in the anticyclone between 370–380K is mainly in the troposphere and the tracer can, indeed, be used for studying cross-tropopause transport. A careful discussion of these points is included now in "Method" and "Appendix".

*2) It would notably improve the paper if a figure similar to Figure 4 but for CO concentrations were included, comparing the model to the observations. Alternatively, a figure similar to the lower panels of Fig. 3 extended to show CO and the origin tracer on additional levels relative to the tropopause (e.g. up to 50 K above the tropopause). Any of these suggested figures would highlight the differences and similarities between CO and the origin tracer (are they due only to the CO lifetime?), provide further confirmation of the correct representation of transport in the model, and facilitate the comparison of the results with previous works.*

The suggested figure showing CLaMS and MLS CO and the anticyclone air mass tracer at levels 10, 20 and 30K above the local tropopause is shown in Fig. 1 of this reply. Overall, the figure is consistent with the discussion in the manuscript. Although the CO anomaly decays somewhat faster in MLS than CLaMS data, the positive anomaly is clear in both datasets up to about 30K above the tropopause, showing

the vertical nature of cross-tropopause transport in the monsoon anticyclone. The anomaly in the inert anticyclone tracer is still clearer at 30K above the tropopause, consistent with our argumentation that the faster attenuation of the CO anomaly is due to the finite CO lifetime. (Note that for the anticyclone tracer simply the mixing ratio is shown to give insight into the respective mass fraction values - the anomaly with respect to the zonal mean, as for CO, would show an even stronger signal in the monsoon). Therefore, also the maps at other levels above the tropopause are consistent with the line of argumentation in the paper. However, as Reviewer 1 recommended publication almost as is, and because we are afraid of overloading the paper with figures and redundant information, we decided not to include this figure into the revised manuscript. But we slightly extended the discussion of Fig. 3.

*3) Figure 2 calls for a more detailed explanation of the large variability in the time series (divided by a factor of 4 in the Figure). In particular, it is unclear to the reader how the standard deviations are defined (with respect to the various years or for each longitude point within each latitudinal band?). This is an important point, because the validity of the conclusions on the seasonality is questioned given the large variability around the time series, and thus it needs to be clarified exactly what the spread represents.*

We agree that this information about the variability shown is necessary for a correct interpretation of the figure. The variability shown is the standard deviation from the zonal averaging, hence the variability for each longitude point at a given latitude at each day. It therefore just shows the large variability in the tracer distribution in mid-latitudes in the zonal direction, which is to a large degree related to Rossby-wave activity. This is characteristic for any long-lived tracer distribution in the lower stratosphere mid-latitudes. We modified the respective text to be clearer on these issues.

Minor comments:

P3L11–18: *A description of the data the temporal and spatial resolution should be included, to help the reader understand how the data is employed for the model comparisons. For example, ACE-FTS data does not allow high-temporally resolved comparison as that in Fig. 4.*

We now included short paragraphs about the vertical/horizontal resolution of the CLaMS model simulation, as well as about the vertical resolution and sampling density of ACE–FTS and MLS in the "Method" section in the revised version.

P3L25: *In Fig. 1d (AMJ) there is little agreement between the model and the observations in the NH. Could you comment on why this might be the case?*

The anticyclone air mass tracer is only initialized during July–August (because enhanced PV gradients for diagnosis of the monsoon anticyclone boundary usually exist only during this period, see appendix). Therefore, the anticyclone tracer present during April–June originates from the monsoon season almost one year ago and has been almost totally flushed out of the NH lower stratosphere. For that reason, no correlation with the enhanced HCN values close to the tropopause (indicating young and polluted air) can be expected. The related (second) paragraph in the "Results" section has been rewritten to clarify this, and also the next comment.

P3L26: *In Fig. 1a there is a maximum of HCN in the tropical pipe, but no corresponding maximum in the origin tracer. Is this associated with the ascent from the year before?*

Yes! The related (second) paragraph in the "Results" section has been rewritten to clarify this (see previous comment).

P3L2–3: *It is not clear what you refer to. Could you refer to specific contours?*

We think you refer to P4 here. The respective paragraph has been rewritten and should be much clearer now.

P3L15: *How do you define the lower edge of the tropical pipe?*

Again, we think you refer to P4 here. We did not diagnose the tropical pipe explicitly and we agree with the Reviewer that identifying the lower edge of the tropical pipe is not straightforward. Many studies show frequent exchange between tropics and mid-latitudes in a layer from the tropopause to about 450 K (e.g., Volk et al., 1996; Konopka et al., 2009; Ploeger et al., 2013), and this layer has been termed for that reason the "tropical transition layer" by Rosenlof et al. (1997). The level considered here (460 K) is just above that layer and we therefore think it should be representative of the lower edge of the tropical pipe. However, as this is not supported by a careful analysis, we reworded the respective sentence to "...460 K (above the layer of frequent exchange between tropics and mid-latitudes, see Rosenlof et al., 1997)...".

P3L20: *I don't see in Fig. 2 that "the anticyclone air exceeds the extra-tropical amount by half a year". Do you mean that from February until the next summer the tropical and extra-tropical monsoon air mass fractions are approximately equal?*

Our explanation here was not clear. What we aimed to say was that the amount of anticyclone tracer in the tropics (at the level where this fraction maximizes) exceeds the anticyclone tracer amount in the extratropics after about half a year (hence from around January on). However, the level of maximum anticyclone air mass fraction in the tropics rises due to tropical upwelling (already visible from Fig. 2 by comparing the timing of maxima for tropical fractions at 400 and 460K). Hence, this higher tropical fraction after January will be located at a level higher than 460K, which was not explicitly shown in the figure. We now included an additional timeseries at 550K in Fig. 2, illustrating this and reworded the respective text to clarify things: "The anticyclone air fraction in the extra-tropical stratosphere peaks with a value that is more than twice as high compared to the tropical anticyclone air fraction. However, the anticyclone air transported to the tropics remains much longer in the stratosphere and exceeds the extra-tropical amount after about half a year (at levels higher than 460K the anticyclone air fraction peaks after January with peak values above the extratropical anticyclone air fraction, see Fig. 2)."

*P4L21–24: As mentioned in the major comment 3), a clarification on how the variability is computed and what it implies for the conclusions on the seasonality is needed.*

See reply to major comment 3.

*P5L11: "cross-tropopause transport into the stratosphere in the monsoon occurs to a large degree in the vertical direction". You could actually quantify to what degree this happens. This is in line with major comment 2), suggesting a more detailed analysis of CO behavior relative to the tropopause, in the model and the observations.*

The anticyclone tracer in the CLaMS model provides quantitative information of how much of the air originates in the monsoon anticyclone. Hence, the clear maximum above the monsoon in the anticyclone tracer maps in Fig. 3 (right column) shows that the major part of the air mass from the anticyclone crosses the tropopause vertically, causing a fraction of approximately 40% of anticyclone air at 10K above the tropopause. Therefore, Fig. 3 already includes quantitative information about cross-tropopause transport (see also the reply to major comment 2).

*P6L20: How is this related to the vertical conduit from the boundary layer?*

Bergman et al. (2013) show that the air masses in the anticyclone at a particular level have been transported through a narrow conduit from levels below. This is qualitatively similar to the regionally confined ("chimney-like") tropopause-crossing of air masses from the anticyclone. However, in our case we consider air masses being transported out of the anticyclone whereas Bergman et al. consider air masses transported into the anticyclone. Furthermore, the vertical coordinates used are different (pressure vs. potential temperature). Therefore, we decided to remove this statement in the revised manuscript version.

Technical corrections/suggestions:

*P1L9: emissions → pollutants*

Changed.

P2L11: *tracer-independent*

Changed.

P2L16: *"and argue that in terms of air mass transport into the stratosphere, the Asian monsoon..."*

Changed.

P2L23–26: *I would recommend introducing the tracer here rather than at the Appendix. Specifically mention that it is defined as unity inside the monsoon anticyclone.*

We agree that the information given in the methods section was not sufficient. We now added a hint at first mention of the tracer ("...We included an air-mass origin tracer ... (see below)..."), and then in the next but one paragraph, where the tracer set-up is explained in more detail, we explicitly state the initialization with unity ("...tracer is initialized with unity...").

P2L31–32: *Here and throughout the paper, PV-value and PV-gradient do not need a hyphen.*

Changed.

Fig. 1: *Perhaps the hatching could be removed, as it is hard to distinguish the colors underneath.*

We already played around a lot with that figure and decided that the hatching is advantageous for distinguishing the locations of highest HCN mixing rations and their correlation with the monsoon model tracer, what is the key message of this plot. Therefore, we would like to keep the figure as is, although the colours become slightly shaded.

Fig. 2: *There is a typo either in the legend or in the figure caption, is the level 450 K or 460 K? Also, the lines are not grey as the caption says, but black.*

The typoe was in the figure caption, the level is 460K – Thanks for pointing this out!

Also the line description in the caption has been corrected to "black".

P5L9: *tropopause-based*

Changed.

P5L29: *Is it really controversial or just a scientific debate?*

We agree that it is more appropriate to call it just a "scientific debate" – Changed.

P6L10: *This does not discard isentropic advection*

Yes, we agree. And it was not our intention to discard it. We only aim at separating cross-isentropic from isentropic transport (not distinguishing isentropic mixing from isentropic advection). We try to be more clear with wording in the revised version by explicitly stating "...horizontal transport (either isentropic advection or mixing...").

P6L16: *emissions ! pollutants*

Changed.

Fig. 5: *Please add a colorbar.*

This figure was aimed to be a schematic and the contour values were chosen to highlight the patterns in the tracer mixing ratio. However, we now provide the contour values in the figure caption.

supplement.pdf

[revised manuscript text omitted]

---

## Author Comment (AC2) · 2 May 2017

We thank the Reviewer for her/his careful consideration of the manuscript and are encouraged by her/his very positive rating of the manuscript. All minor points raised are addressed in the revised version, as described below (Reviewer's comments in italics). Text changes in the manuscript are highlighted in color (except minor wording changes).

Minor remarks:

p.5, l.31: *What is meant with "direct pathway"? I guess quasihorizontal roughly along isentropes from 380K-400K into the tropical tropopause region?*

We agree that our wording here was somewhat too loose. However, to our understanding also Garny and Randel (2016) are not totally clear about the details of their main pathway from the anticyclone to the stratosphere. They write that "the preferred pathway of trajectories is to travel from within the upper-tropospheric anticyclone region to the tropical lower stratosphere (32% of all trajectories). Another 14% are first mixed outside of the anticyclone into the tropical upper troposphere and are subsequently transported upward into the tropical lower stratosphere." As the analysis is neither done in potential temperature nor in tropopause based coordinates, it is difficult to exactly conclude where isentropic mixing occurs. Furthermore, their definition of the tropics covers 30S–45N, such that the anticyclone belongs to the tropics, whereas we differentiate between deep tropics and subtropics (to which we assign the anticyclone). In this sense, our findings regarding cross-tropopause transport in the Asian monsoon are a refinement of Garny and Randel (2016). In the revised verison, we changed the wording to: "Garny and Randel (2016) concluded from 60 day backward trajectory ensembles that the preferred pathway of air masses is to travel from within the upper-tropospheric anticyclone region to the tropical lower stratosphere, but they did not further investigate where (relative to the tropopause) horizontal mixing from the monsoon region to low and high latitudes occurs".

p.5, l.33: *Similarly: "....transported upwards in the monsoon and reaches the extratropical tropopause within a few days...": The monsoon tropopause is meant here with extratropical, please clarify.*

Orbe et al. (2015) state that "Asian air reaches the extratropical tropopause within a few days of leaving the boundary layer and is quasi-horizontally transported into the tropical lower stratosphere...", implying that the monsoon tropopause is considered to be extratropical. Their Fig. 4 shows that the use of pressure as vertical coordinate favors the interpretation of tracer transport from the monsoon towards the NH extratropics, as tracer contours are slanted towards high latitudes. Using potential temperature as vertical coordinate clearly shows cross-tropopause transport in the vertical direction (see our Figs. 3 & 4). Hence, in our opinion considering the monsoon tropopause as

extratropical is arising from the use of pressure as vertical coordinate.

p.5, l.27: *equator wards → equatorwards*

Changed.

Please also note the supplement to this comment:
http://www.atmos-chem-phys-discuss.net/acp-2017-86/acp-2017-86-AC2-
supplement.pdf

[Figure]

**Supplement:**

[revised manuscript text omitted]